



# Introduction and comparison of novel deep learning and optimization approaches to analytical wake modeling of a tilted wind turbine

James Cutler[1], Christopher Bay[2], and Andrew Ning[1]

[1]Brigham Young University, Department of Mechanical Engineering, Provo, UT, 84602
[2]National Renewable Energy Laboratory, National Wind Technology Center, Boulder, CO, 80303

**Correspondence:** James Cutler (asianein@byu.edu)

**Abstract.** This paper introduces innovative optimization and deep learning techniques to enhance the prediction of complex wake dynamics in the downstream wind velocity of tilted wind turbines. Traditional methods for calibrating the Bastankhah wake model often lead to increased errors in wind velocity distribution due to overfitting local wake characteristics. To address this, we propose an additional global optimization step to reduce errors in wind velocity predictions with respect to various

5   wake parameters. Despite this improvement, the Bastankhah model's axisymmetric Gaussian wake shape limits its accuracy for complex wake structures. Therefore, we also propose a deep learning approach, which demonstrates promising results by accurately modeling complex wake shapes across a broader range of tilt angles with minimal computational cost. The deep learning approach achieves near-identical predictions to high-fidelity large-eddy simulations, representing a promising advancement in wake modeling.

## 1   Introduction

As the world continues to pursue renewable energy goals, it is increasingly important to improve the efficiency of wind farms. Floating offshore wind farms have great potential to contribute to the overall renewable energy portfolio; however, they can be costly. The U.S. Department of Energy has a goal to reduce the cost by 70 % by 2035 (U.S. Department of Energy (2022)). One of the main contributors to reduced energy production is the overlapping of upstream turbine wakes over the rotor swept areas

20  of downstream turbines. Wind farms lose between 15% and 20% of energy production for a typical wind farm throughout the year because of wake interference between turbines (Barthelmie et al. (2007); Briggs (2013); Barthelmie et al. (2009);





Barthelmie and Jensen (2010)). To reduce the impact of wake interference, upstream turbine orientation can be coordinated to redirect the wake away from downstream turbines (Kheirabadi and Nagamune (2019)).

Bastankhah and Porté-Agel (2016) developed an analytical wake model that is capable of sufficiently modeling the horizontal deflection in the wake of a yawed turbine using an axisymmetric Gaussian description of the wind speed distribution. However, floating offshore wind turbines can also redirect the wake vertically based on the platform tilt (Wisatesajja et al. (2019)). Additionally, a downward deflected wake interacts with the ground and can begin to exhibit more complex behavior (Johlas et al. (2022)). A key challenge in enhancing the efficiency of floating offshore wind farms is developing and refining wake modeling to accurately capture more complex wake dynamics such as tilted wind turbine wakes. This study introduces an improvement to current analytic wake modeling practice as well as a novel deep learning approach to modeling complex wake dynamics. The improvements and limitations of these approaches are demonstrated in the modeling of a tilted wind turbine's wake for various tilt angles.

The Bastankhah wake model was derived by applying conservation of mass and momentum to a Gaussian distribution description of the wake velocity deficit (Bastankhah and Porté-Agel (2014)). This approach to wake modeling can accurately predict the far wake for varying wind speed, turbulence intensity, and rotor size. The main assumption of the model is that the wake maintains a Gaussian description in both the horizontal and vertical velocity distributions. However, for large deflections in the wake trajectory, the wake begins to form a kidney bean shape, and therefore the Bastankhah wake model is only applicable to a limited range of wake deflection (Bastankhah and Porté-Agel (2016)). For horizontal deflection, this limited range is sufficient for modeling yaw angles that would be applied in wind farm control strategies (Bastankhah and Porté-Agel (2019)). However, when tilted turbines deflect the wake downward, the interactions with the ground deform the wake significantly more than yawed turbines. Thus, the range of tilt angles that the Bastankhah wake model can accurately model are smaller than the range of yaw angles. Additionally, the effect of the ground is not accounted for in the application of conservation of mass and momentum in the original derivation of the Bastankhah wake model.

Nanos et al. (2020) observed wake shape and deflection for tilted turbines using particle image velocimetry (PIV) calibrated computational fluid dynamics (CFD) simulations. Their results show that when a turbine is deflected toward the ground, the wake compresses vertically as it expands horizontally. Thus, to account for the effect of the ground, the definition of wake expansion in the Bastankhah wake model must account for vertical compression and horizontal expansion with increasing tilt angle. The approach used to account for yaw in the Bastankhah wake model can be applied to also account for tilt (Bastankhah and Porté-Agel (2016)). This approach involves analyzing the patterns of wake growth and deflection in high-fidelity simulations of tilted turbines. When analyzing yaw, there is an assumed insignificant deflection in the vertical direction; thus, the analysis of wake growth and deflection can be based on stream-wise slices of the flow field. However, for tilt, there is a significant amount of both vertical and horizontal deflection as the wake approaches the ground (Porté-Agel et al. (2020)). Therefore, in this study, we analyze and define deflection and wake growth based on cross-stream slices of the stream-wise velocity at varying downstream locations. We used SOWFA (Simulator fOr Wind Farm Applications) to simulate the wake of a 5-MW National Renewable Energy Laboratory (NREL) reference turbine over varying fixed tilt angles (Churchfield et al. (2012)). Based on the analysis of cross-stream slices, we can define a range of tilt angles that hold the assumption of a Gaussian



distribution description as well as any necessary additional empirical relationships for variables dependent on tilt. Then local optimizations can be conducted to reduce the error of the wake growth and deflection predictions with respect to the parameters of the empirical relationships. Although constrained to a limited range of tilt angles, this approach can define and calibrate the necessary additions and adjustments to the Bastankhah wake model to account for tilt.

To improve upon this approach, this study introduces a novel additional step of optimization that significantly improves the accuracy of the wind distribution prediction. The main objective of wake modeling is to accurately predict the wind speed distribution. However, current wake modeling approaches focus on various local optimizations in order to accurately predict wake growth and wake deflection. This can lead to over-fitting the relationships that define wake growth and deflection (Farajpour and Atamturktur (2012); Li et al. (2016)). Thus, after defining the analytical and empirical relationships of the wake model, an additional optimization step can be conducted to reduce the root mean squared error between the SOWFA velocity field and the model predictions with respect to the local parameters that define wake growth and deflection.

The additions and adjustments defined in this study are not generalizable across varying turbine types. However, the modeling and optimization approach described in this paper will allow these and other modifications to be more generalizable, provided there is sufficient data. However, even with more data, the Bastankhah wake model is incapable of modeling the kidney bean shaped wake that occurs at large tilt angles; therefore, the additions and adjustments are only valid for platform tilt angles less than 15° (Churchfield et al. (2016)).

Therefore, in this study we additionally demonstrate a novel deep learning approach that can develop a model capable of handling complex wake structures and large tilt angles. The objective of the optimization within the deep learning approach is the same as the suggested additional optimization step, except the neural network can model complex wake structures without the time typically required to quantify the wake deflection and growth (Ti et al. (2020)).

Recently, applying deep learning to wake modeling has become an important topic of study due to the capability of deep learning to model high-fidelity wake characteristics at a fraction of the computational cost (Zhang and Zhao (2020); Ti et al. (2020); Pawar et al. (2022)). A majority of these approaches involve multi-fidelity deep learning where the results of the Bastankhah wake model are correlated to high-fidelity simulation results. These approaches demonstrate the ability to train a neural net to generate accurate high-fidelity results. However, these approaches are limited to learning a fixed stream-wise slice of data, which can only be applied to determine a vertical or horizontal velocity profile of the wake at a downstream turbine. In this study, the proposed deep learning approach learns to predict the cross-stream slice of the wake of a tilted turbine at any downstream distance for any tilt angle, rather than a stream-wise slice. Thus, this deep learning approach is able to resolve the wake in three-dimensional space for any tilt angle of an upstream NREL 5-MW turbine. This approach provides the velocity predictions necessary to determine accurate overlap of a tilted turbine wake with any downstream turbine.

The paper is organized as follows. The observations, analysis, and definitions of key parameters in modeling tilted turbine wakes for the optimization and deep learning approach are detailed in Sect. 2. The calibration and optimization of model additions and adjustments as well as the training of the deep neural net are detailed in Sect. 3. Comparison of the predicted velocity field with the various modeling techniques is detailed in Sect. 4. A summary of results and future work is presented in Sect. 5.





## 2 Analysis of tilted turbine wakes in large-eddy simulations

The first step in determining the appropriate modifications and additions to the Bastankhah wake model is observing the behavior of the tilted wind turbine wake. Then the variables and empirical relationships necessary to model wake growth and deflection can be defined and calibrated. In this initial stage of analyzing the high-fidelity SOWFA data, the velocity field was compiled into training data for both the additional optimization and deep learning approaches. The training data consists of cross-stream slices of the velocity field at several downstream distances (see Fig. 1).

### 2.1 Tilted turbine wake analysis

A 5-MW NREL reference turbine was simulated in SOWFA over varying degrees of tilt at a wind speed of 8 m/s and a low turbulence intensity (Churchfield et al. (2012)). A majority of the simulations were focused on positive tilt in order to thoroughly analyze the effect of the ground on the wake. Overall, the results of the SOWFA simulations confirm similar trends to previous studies of tilted turbines. When the wake approaches the ground, it expands horizontally and compresses vertically (see Fig. 1a), and when the wake is directed upward, it stretches vertically and gradually expands horizontally (see Fig. 1b) (Johlas et al. (2022); Annoni et al. (2017)).

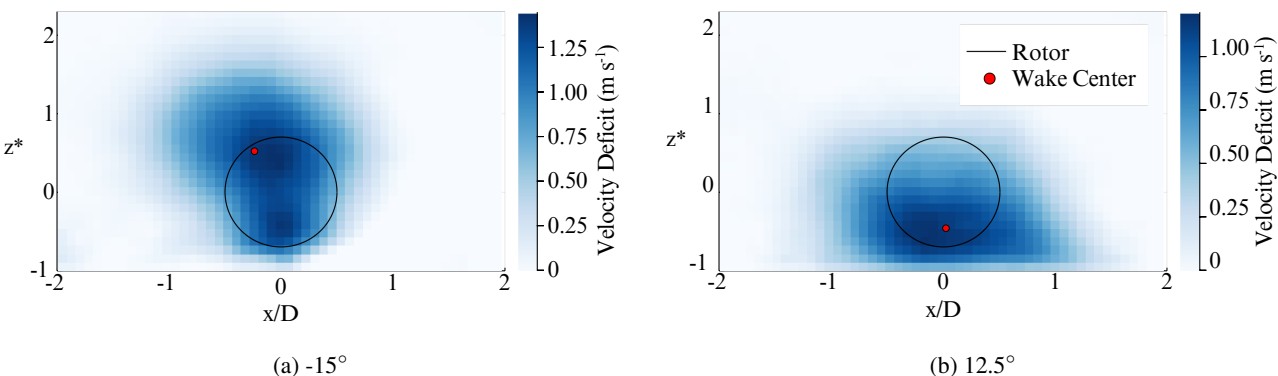

(a) -15°                                           (b) 12.5°

**Figure 1.** Cross-stream slice of the velocity deficit at 12 x/D downstream of a turbine tilted -15°(a) and 12.5°(b). z* represents the vertical position normalized with the hub height (90 meters).

A common method for analyzing wake deflection and expansion involves observing the stream-wise velocity profiles. These profiles are obtained from vertical and horizontal slices of the velocity field, taken at the center of the turbine rotor (Bastankhah and Porté-Agel (2016); Annoni et al. (2017)) (see Fig. 2a). Although this method may suffice for small tilt and yaw angles (Fleming et al. (2014)), there is a significant amount of horizontal and vertical deflection in the wake for larger tilt angles (see Fig. 1a).



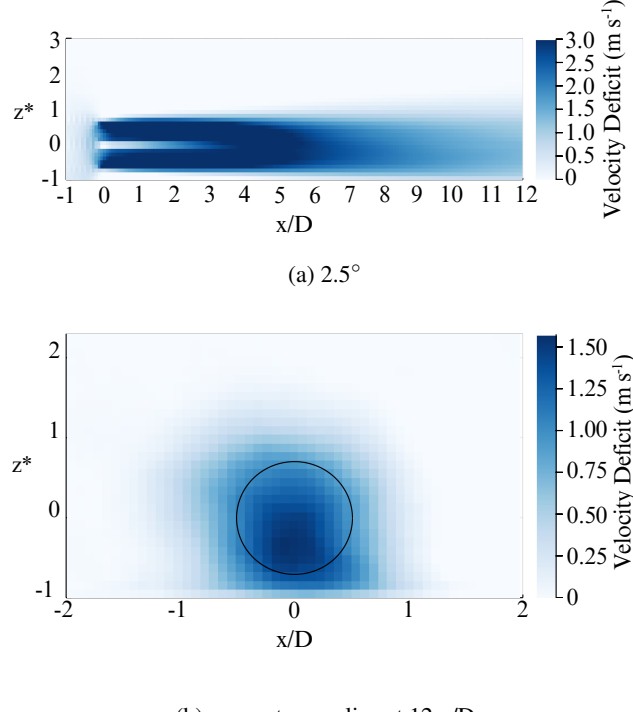

(a) 2.5°

(b) cross-stream slice at 12 x/D

**Figure 2.** Stream-wise slice of the velocity deficit centered at a 2.5° tilted turbine (a) and a cross-stream slice at 12 x/D (b). z* represents the vertical position normalized with the hub height (90 meters).

The observed vertical deflection of the wake can be inaccurately estimated when pulled from the stream-wise slice of the flow field. There are more reasonable and accurate results when the vertical deflection is based on the true wake center from the cross-stream slices. When comparing the vertical deflection of the wake estimated from stream-wise slices and cross-stream slices, it is evident that analysis of a stream-wise slice leads to significant inaccuracies (see Fig. 3). Observing downstream vertical velocity profiles based on a stream-wise slice of the flow field can be misleading due the horizontal deflection displacing

the true center of the wake out of the stream-wise plane (see Fig. 1a).





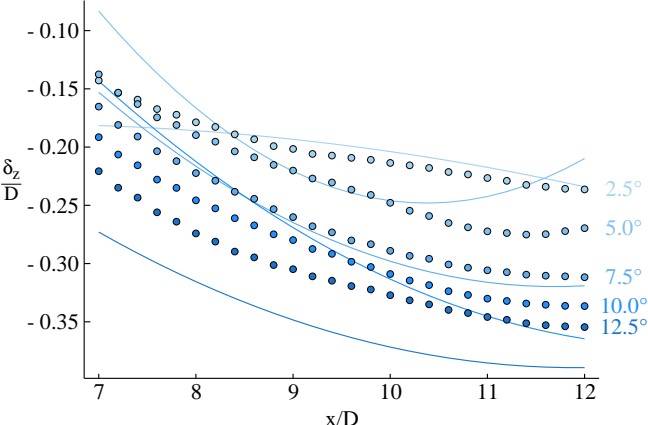

**Figure 3.** Deflection of tilted turbine wakes as observed from cross-stream slices in SOWFA (marked with points) and the deflection observed from a stream-wise slice (marked with solid lines).

For this analysis, the vertical and horizontal velocity profiles are based on the point of maximum velocity deficit at each observed cross-stream section of the wake (see Fig. 2b). For each vertical and horizontal velocity profile a normal Gaussian fit was used to determine the standard deviations ($\sigma_z$ and $\sigma_y$ respectively). In the Bastankhah wake model, the standard deviations are a measure of the wake width in order to define the growth of the wake as it moves downstream (Bastankhah and Porté-Agel (2016)). Equations 1 and 2 define $\sigma_y$ and $\sigma_z$ as having a linear relationship with the downstream distance ($\frac{x-x_0}{d}$) where the slope ($k_y$ and $k_z$) is determined by applying a linear fit to $\sigma_y$ and $\sigma_z$. In the Bastankhah wake model, $\sigma_y$ and $\sigma_z$ are used as inputs for the velocity deficit ($\frac{\triangle\hat{U}}{\hat{U}_\infty}$) (see Eq. 3). These relationships and equations originate from the original Bastankhah wake model derivation where $C_T$ is the coefficient of thrust, $\gamma$ is turbine tilt, $D$ is the rotor diameter, $y_h$ is the hub-height, $\alpha$ is the veer in the incoming wind distribution, and $\delta_y$ and $\delta_z$ are the horizontal and vertical deflections respectively (Bastankhah and Porté-Agel (2016)).

$$\frac{\sigma_y}{D} = k_y \frac{x-x_0}{D} + \sigma_{y0} \tag{1}$$

$$\frac{\sigma_z}{D} = k_z \frac{x-x_0}{D} + \sigma_{z0} \tag{2}$$

$$\frac{\triangle\hat{U}}{\hat{U}_\infty} = \left(1 - \sqrt{1 - \frac{C_T\cos(\gamma)}{8(\sigma_y\sigma_z/D^2)}}\right)\exp\left[-0.5\left(\frac{y-y_h+\delta_y+x\tan(\alpha)}{\sigma_y}\right)^2\right]\exp\left[-0.5\left(\frac{z-z_h-\delta_z}{\sigma_z}\right)^2\right] \tag{3}$$

### 2.1.1 Skewed gaussian fit

The main foundational assumption in the Bastankhah wake model assumes a normal Gaussian shape in the vertical and horizontal velocity profiles. However, when the wake compresses vertically it forms a skewed Gaussian shape (see Fig. 4). Skewed





Gaussian shapes have also been observed in analyses of yawed turbine wakes; however, the skew was insignificant enough

to maintain the assumption of a normal Gaussian shape (Bastankhah and Porté-Agel (2016)). Similarly, for small positive tilt

angles, the skew is negligible and a normal Gaussian fit sufficiently defines the wake shape and wake growth. However, with

the presence of the ground, the skew can not be ignored for large tilt angles. A skewed Gaussian fit would be better suited to

approximate the vertical velocity profile; however, this would conflict with assumptions used to derive the Bastankhah wake

model (Bastankhah and Porté-Agel (2016)). Although the skew becomes more prominent for large tilt angles, the deflection of

the center of the wake places the bottom portion of the wake away from the rotor swept areas of downstream turbines (assuming

the same hub height). Therefore, a normal Gaussian fit can still be used as long as it accurately approximates the upper portion

of the vertical velocity profile (see Fig. 4).

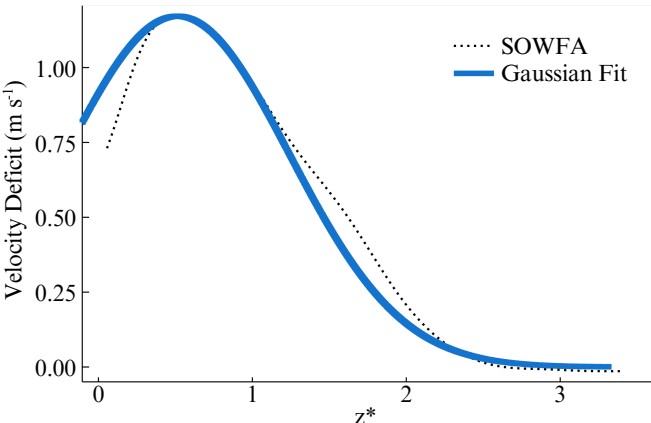

**Figure 4.** Vertical velocity profile at 14 x/D downstream of a turbine with 12.5° of tilt.

In order to focus on accurately estimating the upper portion of the vertical velocity profile the profile is split at the point of

max velocity deficit. Then the upper portion of the velocity profile, is mirrored across the point. This forms a normal Gaussian

shape where a normal Gaussian fit is used to find $\sigma_z$. In order to accurately define the relationship between $\sigma_z$ and tilt, SOWFA

simulations were run for a single turbine at tilt angles of 2.5°, 5°, 7.5°, 10°, and 12.5° (see Fig. 5a). A normal Gaussian fit

without any required mirroring of the data was used to measure $\sigma_y$ (see Fig. 5b). Similar to what has been observed with

turbine yaw, $\sigma_z$ and $\sigma_y$ can be observed to increase linearly with respect to the downstream distance even for larger angles

of tilt. Bastankhah and Porté-Agel (2016) observed that the rate at which $\sigma_y$ increased was constant over varying yaw angle.

However, the rate at which $\sigma_z$ and $\sigma_y$ increase is variable over varying tilt angles. Thus, empirical relationships are necessary

to define the change in slope over varying tilt angles for both $\sigma_z$ and $\sigma_y$.

### 2.1.2 Kidney bean shape

When the turbine is tilted beyond 15°, a kidney bean shape begins to form (see Fig. 6). The kidney bean shape would require

a double Gaussian shape to approximate, whereas the Bastankhah wake model derivation relies on a single Gaussian shape

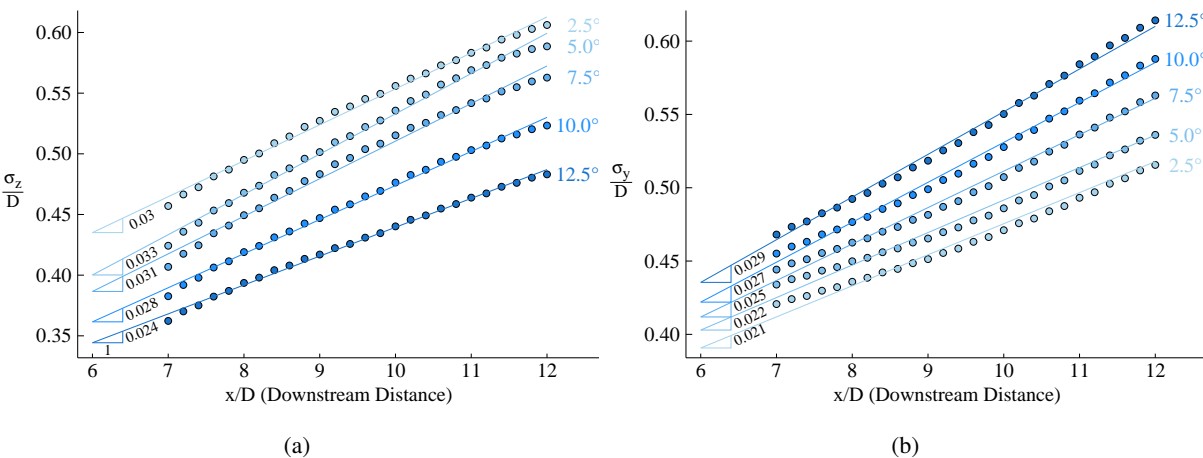

|           |           |
|-----------|-----------|
| (a)       | (b)       |

**Figure 5.** $\sigma_z$ for the mirrored upper portion of the vertical velocity profile (a) and $\sigma_y$ (b) measured over a range of turbine tilt angles and downstream distances.

description (Johlas et al. (2022)). Therefore, the calibrated analytical wake model described in this paper is bounded to tilt angles less than $15°$. This limit is reasonable as it is in line with limits set for fixed offshore platform tilt (Ramachandran et al. (2017).

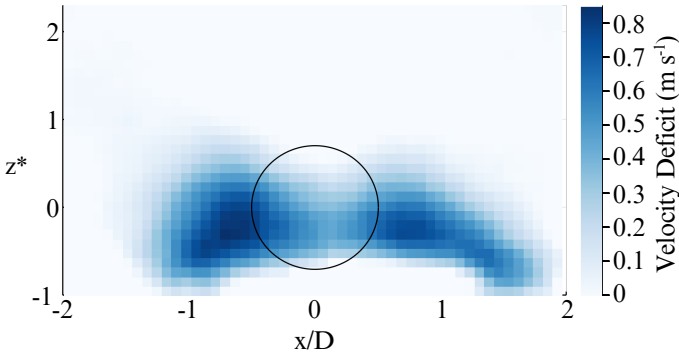

**Figure 6.** Velocity deficit from SOWFA simulation at 14 x/D downstream of a turbine tilted $25°$.

### 2.1.3 Positive tilt deflection

The deflection term in the original derivation of the Bastankhah wake model is dependent on $\sigma_z$ and $\sigma_y$; however, when
implementing $\sigma_z$ and $\sigma_y$ from Figs. 5a-5b, the vertical deflection does not come close to matching the deflection observed in the SOWFA simulations, presumably due to wake interactions with the ground (see Fig. 7).





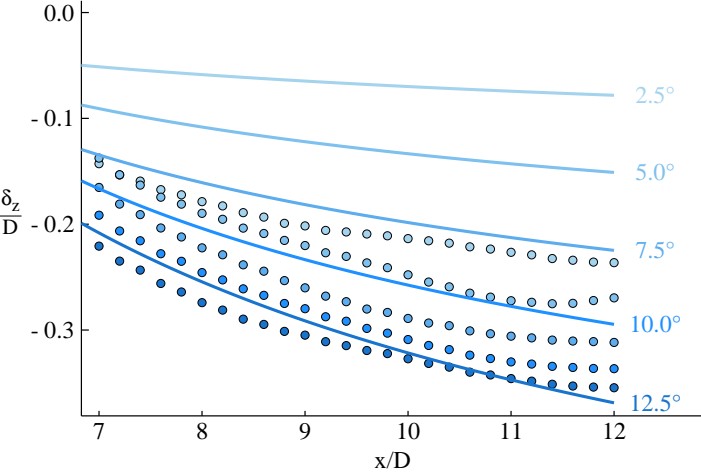

**Figure 7.** Deflection of tilted turbine wakes as observed in the SOWFA data (marked with points) and the deflection predictions of the Bastankhah wake model (marked with solid lines).

Therefore, a surrogate model is utilized to define vertical deflection as a function of tilt (see Eq. 4-5). The surrogate model was chosen such that a simple linear least squares solution could be used to solve for the coefficients analytically and accurately (Martins and Ning (2021)) (see Eq. 6-7). Figure 8 shows the calibrated surrogate model results (marked with solid lines) and the SOWFA data points used for calibration in the vertical (on the left) and horizontal (on the right) directions.

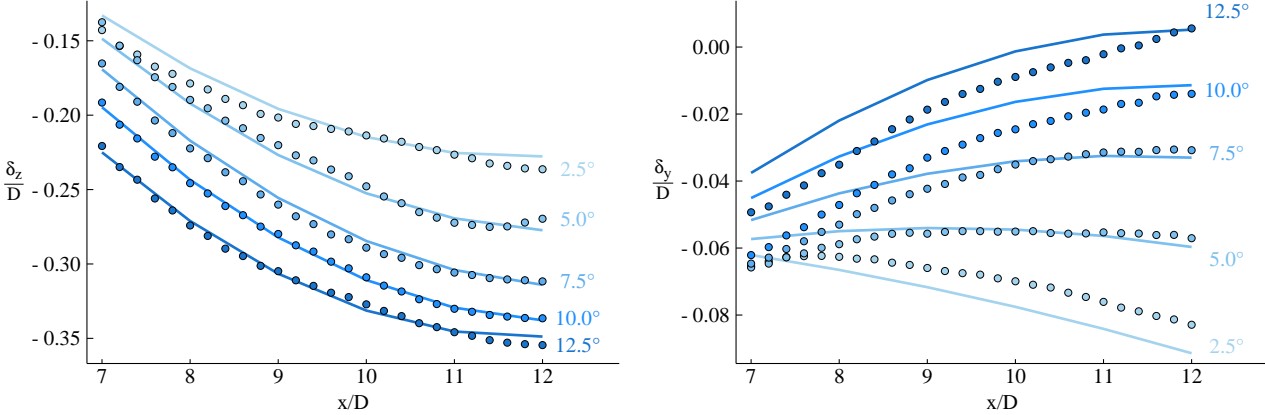

**Figure 8.** Deflection of tilted turbine wakes as observed in the SOWFA data (marked with points) and the deflection predictions of the calibrated surrogate deflection models (marked with solid lines).

$$\frac{\delta_z}{D} = c_1\gamma + c_2\gamma^2 + + c_3\frac{x}{D} + c_4\left(\frac{x}{D}\right)^2 + c_5\frac{x}{D}\gamma + c_6\gamma^2\frac{x}{D} + c_7\left(\frac{x}{D}\right)^2\gamma + c_8 \qquad (4)$$



$$\frac{\delta_y}{D} = d_1\gamma + d_2\gamma^2 + + d_3\frac{x}{D} + d_4\frac{x}{D}\gamma + d_5\gamma^2\frac{x}{D} + d_6\left(\frac{x}{D}\right)^2\gamma + d_7 \tag{5}$$

$$c_1, c_2, ..., c_8 = 2.0921, -7.9725, -0.0854, 0.0041, -0.3663, 0.9701, 0.0045, 0.2840 \tag{6}$$

$$d_1, d_2, ..., d_7 = -1.7558, 2.4323, -0.0125, 0.3187, -0.3131, -0.0081, 0.0212 \tag{7}$$

Since this deflection surrogate model is calibrated to match the behavior of the NREL 5-MW wind turbine, it may be used for turbines with a similar hub height to rotor diameter ratio (0.75). However, for turbines with significantly different hub height to rotor diameter ratios, the deflection surrogate will need to be recalibrated. This can be fast and accurate due to the simplicity of the surrogate model chosen.

This analysis has shown that in order for the Bastankhah wake model to be able to predict the behavior of a tilted turbine wake, the deflection definition requires a replacement surrogate model and $\sigma_y$ and $\sigma_z$ need additional empirical relationships in order to define their dependence on tilt.

## 3 Model calibrations

To account for tilt in the Bastankhah wake model, the dependence of wake growth on tilt is defined by determining the appropriate empirical models and fitting them to the observed patterns of wake growth. To further improve the accuracy of the modified Bastankhah wake model, a simple additional step of optimization is implemented where the main objective is defined to minimize the root mean square (RMS) error between the SOWFA data velocity field and the model predictions with respect to the coefficients of the additional empirical relationships. For the deep learning approach, the same objective of the additional step of optimization is maintained, but instead with respect to the weights of a neural net. This allows the optimization to explore asymmetric wake shapes outside the constraints of the Bastankhah wake model in order to further reduce the RMS error.

### 3.1 Empirical definitions

In studies of horizontal wake deflection, $k_y$ and $k_z$ (see Eqs. 1-2) have been observed to be constant across varying yaw angles (Bastankhah and Porté-Agel (2016)). However, $k_y$ and $k_z$ change significantly enough across varying tilt angles to assume dependence (see Fig. 9). As the turbine's tilt increases positively, $k_y$ increases and $k_z$ of the upper portion of the wake decreases. To define the dependence of $k_z$ and $k_y$ on $\gamma$, local optimizations are implemented to fit a parabolic and linear definition to $k_z$ and $k_y$ data, respectively (see Fig. 9).

The decreasing $k_z$ for the mirrored upper portion of the wake would eventually result in negative $k_z$ and therefore eventually define $\sigma_z$ to be negative, resulting in a square root of a negative value in the velocity deficit equation (see Eq. 2). However, this





would be well out of the range of reasonable tilt angles. For comprehension, Fig. 9 displays the tilt angle in degrees; however,

195  $\gamma$ is defined in radians in the relationship for $k_y$ and $k_z$.

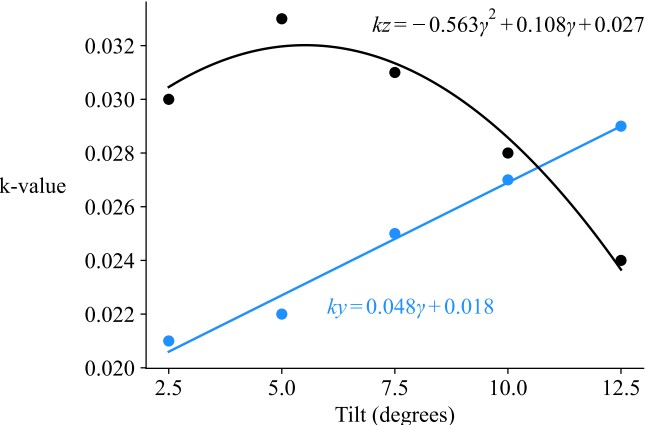

**Figure 9.** Empirical relationships between $k_z$ and $k_y$ and turbine tilt ($\gamma$).

The change in $\sigma_{z0}$ across varying tilt angles is also significant enough to require a definition for $\sigma_{z0}$ as a function of tilt (see Fig. 10). In previous studies of wake deflection, $\sigma_{y0}$ has been assumed to be constant at around $0.354$; however, for our analysis, Fig. 10 shows that $\sigma_{y0}$ is constant at around $0.266$ (Bastankhah and Porté-Agel (2016)). Figure 10 also shows that as tilt increases, $\sigma_{z0}$ of the upper portion of the wake begins to converge at around $0.2$. In order to define the dependence of $\sigma_{z0}$

200  on tilt, a local optimization is conducted to fit a logarithmic definition to the $\sigma_{z0}$ data. Again, it is important to note that the definitions for $k_z$, $k_y$, $\sigma_{y0}$, and $\sigma_{z0}$ are functions of tilt in radians.

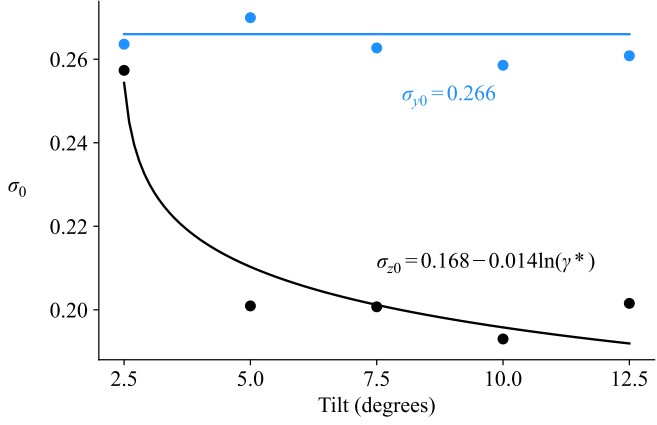

**Figure 10.** $\sigma_{z0}$ and $\sigma_{y0}$ for varying angles of turbine tilt. Note that $\gamma^* = \gamma - 0.0419$.





## 3.2 Additional optimization

In order to further improve the accuracy of this modified Bastankhah wake model, an additional optimization is conducted to minimize the RMS error between the SOWFA data velocity field and the model predictions with respect to the parameters that define the relationship between wake shape and $\gamma$ (see Eq. 8).

$$
\begin{aligned}
&\text{minimize} && \text{RMS} \\
&\text{w.r.t.} && b_1, b_2, b_3, b_4, b_5, b_6, b_7, b_8, b_9, b_{10} \\
&\text{subject to} && 0.35 \leq \sigma_z, \sigma_y \leq 1.0
\end{aligned}
\tag{8}
$$

$\sigma_y$ and $\sigma_z$ are functions of $\gamma$ through the definitions of $k_y$, $k_z$, $\sigma_{z0}$, and $\sigma_{y0}$. From Figs. 9 and 10, we can define $k_y$, $k_z$, $\sigma_{z0}$, and $\sigma_{y0}$ with the following:

$$k_z = b_1 \gamma^2 + b_2 \gamma + b_3 \tag{9}$$

$$k_y = b_4 \gamma + b_5 \tag{10}$$

$$\sigma_{z0} = b_6 - b_7 \ln(\gamma - b_8) \tag{11}$$

$$\sigma_{y0} = b_9 \tag{12}$$

$$\alpha = b_{10} \tag{13}$$

The coefficients $b_1$ to $b_9$ define $\sigma_y$ and $\sigma_z$, and $b_{10}$ represents $\alpha$, the incoming wind angle, which is a measurement of the change in incoming wind angle over the rotor swept area of the tilted turbine. The incoming wind angle is used in Eq. 3 to account for veer in the wake shape and is kept constant over varying tilt angles, assuming that there is no significant difference in $\alpha$ over varying $\gamma$. The coefficients $b_1$ to $b_{10}$ are the variables that the optimization modifies in order to reduce the error between the model predictions and SOWFA (see Eq. 8). The model predictions are defined by Eq. 3, where the velocity deficit ($\Delta U$) is a function of $\gamma$, $\sigma_y$, $\sigma_z$, $\alpha$, $\delta_z$, and $\delta_y$. The coefficients for $\delta_z$ and $\delta_y$ are not modified in the optimization because deflection is a physical measurement that accurately defines the trajectory of the wake center as opposed to $\sigma_y$ and $\sigma_z$, which are estimates of the wake's general shape.



The definitions of $\sigma_y$ and $\sigma_z$ are based on measurements that are highly dependent on the ability of a normal Gaussian fit to accurately define the horizontal velocity profile and the upper portion of the vertical velocity profile. Therefore, $\sigma_y$ and $\sigma_z$ are rough estimates of the expansion of the wake in the horizontal and vertical directions, respectively, and as such should not be treated as concrete measurements. Results from observations of $\sigma_y$ and $\sigma_z$ are used to define the empirical relationships

between wake shape and $\gamma$. Then the additional optimization step can fine-tune the coefficients to further reduce the RMS error of the wake model's flow field predictions.

The objective of the optimization is defined as the normalized sum of the RMS error of the difference between a vertical 2D velocity slice ($U$) from SOWFA and the model prediction of the same area ($U_m$) over varying tilt angles ($\gamma$) and downstream distances (x/D). The RMS portion of the objective is defined in Eq. 14, where $N_y$ and $N_z$ are the total number of points in the

horizontal and vertical directions over the cross-stream velocity slice, respectively (see Fig. 1b for an example of the cross-stream velocity slice area). The objective for the optimization is defined in Eq. 15, where $N_{x/D}$ and $N_\gamma$ represent the total number of downstream distances and tilt angles used in the optimization. The span of the tilt angles and downstream distances are defined in Eqs. 16 and 17.

In order to guide the optimization to reach reasonable results, $\sigma_z$ and $\sigma_y$ are constrained to be greater than 0.35 and less than

1.0 based on observations in Figs 5a and 5b.

$$\text{RMS}_{ij} = \sqrt{\frac{\sum_{k=1}^{N_y} \sum_{l=1}^{N_z} |\Delta \hat{U}_{kl}^2 - \Delta \hat{U}m_{kl}^2|^2}{N_y N_z}} \tag{14}$$

$$\text{RMS} = \frac{\sum_{i=1}^{N_{x/D}} \sum_{j=1}^{N_\gamma} \text{RMS}_{ij}}{N_{x/D} N_\gamma} \tag{15}$$

$$\gamma = 2.5°,\ 5.0°,\ 7.5°,\ 10.0°,\ 12.5° \tag{16}$$

$$\text{x/D} = 7.0,\ 8.0,\ 9.0,\ 10.0,\ 11.0,\ 12.0 \tag{17}$$

### 3.2.1 Optimization results

When comparing the RMS error of the original locally optimized coefficients, to that of the additional optimization coefficients there was an overall reduction in RMS error by about 17% (see Fig. 11). This is a significant reduction in the RMS error that only requires a simple additional step of optimization.



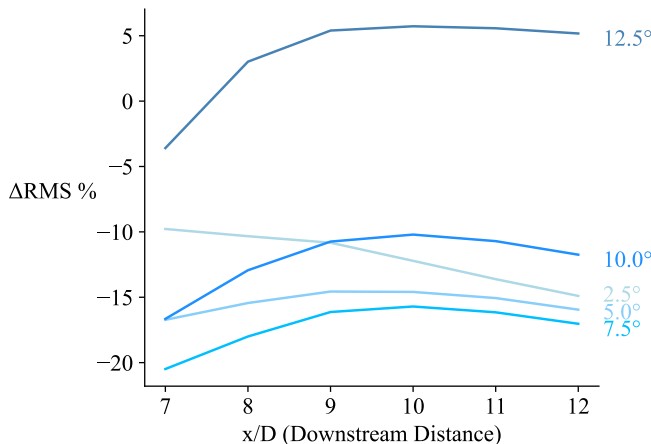

**Figure 11.** Percent difference between the RMS error with the original coefficients and the RMS error with optimized coefficients.

Overall, there was a 10% to 20% reduction in the RMS error for tilt angles ranging from 2.5° to 10.0° (see Fig. 11). However,
beyond 10.0° there was an increase in RMS error by about 5%. This could be due to the normal Gaussian shape assumption
being unable to accurately capture the deformation in the wake shape at 12.5° of tilt due to ground effect. Therefore, another
benefit of this additional optimization step is being able to test the limits of the model. Initial observations suggested the wake
began to form a kidney bean shape at around 15°. However, the optimizer could only reduce the overall RMS error by reducing
the RMS error from 2.5° to 10.0°, while increasing the RMS error of 12.5° tilt. This suggests that this modified Bastankhah
wake model is limited in accuracy for 12.5° of tilt. The limited accuracy is potentially due to the wake beginning to form a
slight double Gaussian shape. This additional optimization step helps identify a limit for allowable tilt angles. Initially, the
upper limit was estimated to be 15°; however, from the optimization, the limit should be around 10°.

In terms of analyzing what specifically the optimizer adjusted in $\sigma_y$ and $\sigma_z$, Fig. 12 reveals that $k_y$ stayed essentially the
same whereas $k_z$ was adjusted significantly in order to reduce the RMS error. The decrease in $k_z$ means the optimizer reduced
the RMS error by reducing the vertical stretching of the wake (see Fig. 13a). When observing only the SOWFA data, $k_z$ was
observed to rapidly decrease as the tilt increased; however, the optimizer found greater accuracy in defining $k_z$ as converging
to a value of 0.025.





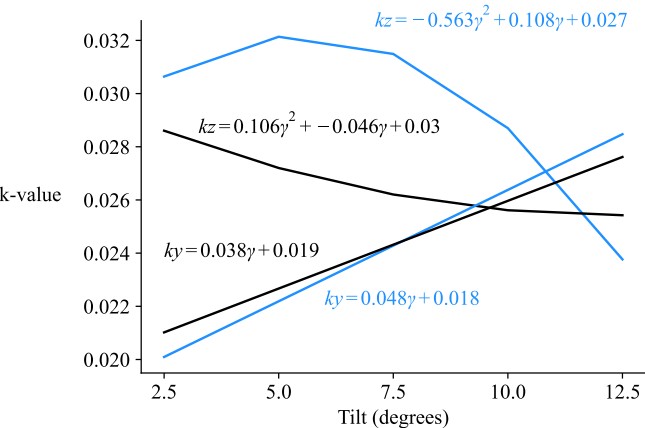

**Figure 12.** $k_y$ and $k_z$ defined by the locally optimized (blue) and the additional optimization (black) coefficients.

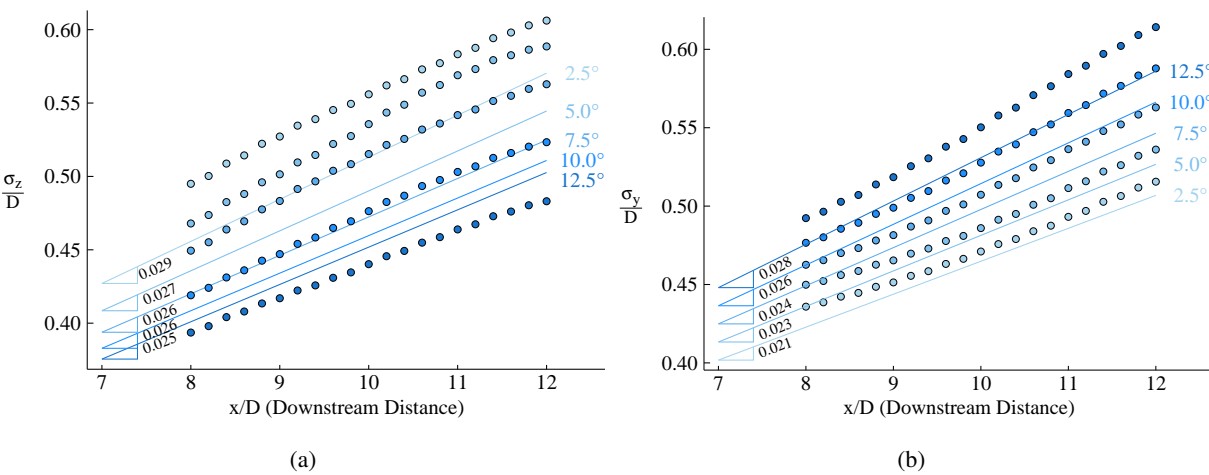

**Figure 13.** Measured $\sigma_z$ and $\sigma_y$ (marked with points) and optimized $\sigma_z$ and $\sigma_y$ (marked with lines) over varying tilt angles and downstream distances.

Figure 13b shows that $\sigma_y$ was also reduced, which means the horizontal expansion was lessened as the wake moved downstream. The overestimate in the measurements of $\sigma_y$ and $\sigma_z$ reveals a potential bias in our initial analysis. The results for $\sigma_{y0}$ and $\sigma_{z0}$ were essentially the same as what had initially been defined, with a $\sigma_{y0}$ of 0.255 and a decreasing $\sigma_{z0}$ over increasing tilt angle (see Fig. 14).





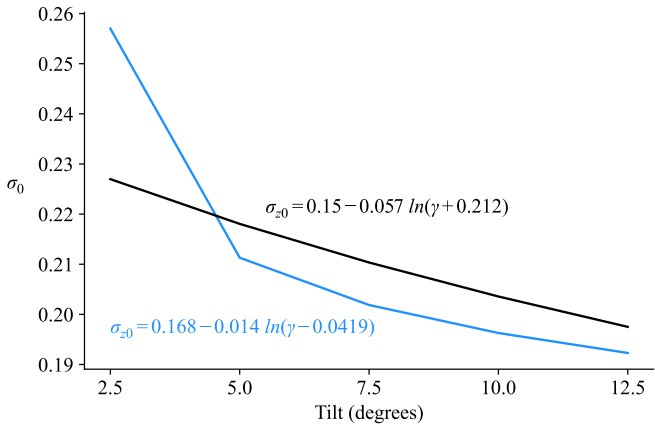

**Figure 14.** $\sigma_{z0}$ defined by the locally optimized coefficients (marked with blue) and the additional optimization coefficients (marked with black).

The additional step of optimization helps identify the limits of the modified Bastankhah wake model as well as improve the overall accuracy of its predictions. Therefore, considering the simplicity of this additional step, it is beneficial to include in future analytical wake modeling endeavors.

### 270    3.3    Deep learning training

Although the additional step of optimization significantly reduced the RMS error of the modified Bastankhah wake model, the model became limited to a small range of feasible tilt angles and required extensive analysis. To improve the modified Bastankhah wake model, there would need to be significantly more analysis of varying tilt angles over varying wind speeds and hub heights. However, this could take a significant amount of time and would continue to remain limited to a small

range of tilt angles. The most extensive portion of this process comprised analyzing and defining the relationships between tilt, deflection, and wake growth. Therefore, implementation of a deep learning approach could remove the need to define deflection and wake growth. Instead, a neural net can learn to define the relationship between tilt and a downstream cross-stream slice of the flow field. Additionally, a deep learning approach would be unhindered by the limitations of assuming a Gaussian wake shape and therefore capable of modeling complex wake features when the wake interacts with the ground.

In this study we implemented a simple neural net with four layers. After each layer, we normalized the batch and used rectified linear unit (ReLU) activation functions. The four layers take in the tilt angle and downstream distance, then expand the size of the first layer from 2 to 128, 128 to 256, 256 to 512, and finally 512 to an image that is 31x51 pixels (SOWFA data resolution for cross-stream slice).

   In order to thoroughly train our neural net, we used 1,850 cross-stream slices from the SOWFA data velocity field over vary-

ing tilt angles ranging from -35° to 25°. The 1,850 images were then randomized into separate training and validation datasets using PyTorch's randperm function, which implements the Mersenne Twister pseudorandom number generator (Imambi et al. (2021)).





Our deep learning approach involves two loss functions, SSIM (Structural Similarity Index) and RMS error (Shrestha and Mahmood (2019)). The RMS error is defined the same as in Eq. 14, except that it only compares one cross-stream slice at a
time. The SSIM objective is used to initially train the neural net in order to remove noise and capture the general shape of the wake (Nilsson and Akenine-Möller (2020)). The SSIM objective alone results in an overly averaged version of the wake due to the Gaussian averaging in the SSIM objective. Thus, additional training with the RMS error objective allows the neural net to further generate finer structures in the wake. Our training consisted of 2,500 epochs of training with the SSIM objective followed by 2,000 epochs of training with the RMS error objective (see Fig. 15).

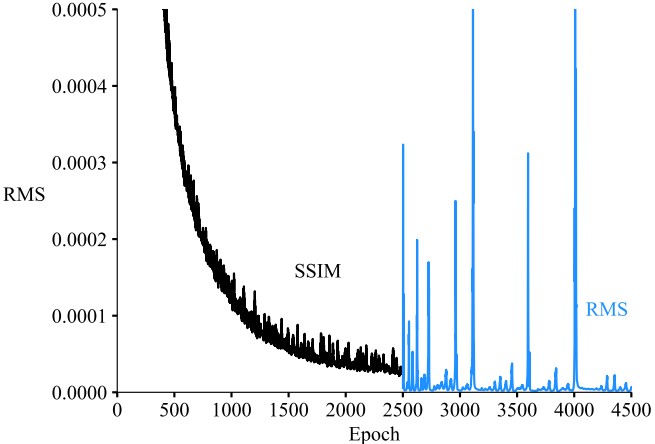

**Figure 15.** Reduction in RMS error of the generated image and SOWFA over 2500 epochs with an SSIM objective followed by 2000 epochs with an RMS error objective.

This approach resulted in a nearly 95% reduction in the RMS error from the optimized modified Bastankhah wake model. Figure 16 demonstrates that there is no significant difference between the generated cross-stream slice of the wake (see Fig. 16b) and the SOWFA data (see Fig. 16a) for a turbine tilted -20°. The main inaccuracy is near the center of the wake, with an overestimate of the velocity deficit by about 0.15 m/s (see Fig. 16c). However, the inaccuracy is insignificant, with a maximum RMS error of about 5% localized at the center of the wake. With this simple neural net, the power of a downstream turbine could be
estimated as accurately as a SOWFA simulation at a fraction of the time it takes to run a SOWFA simulation. Depending on computing power and resources, a SOWFA simulation of one tilted turbine can take up to 10 hours, whereas this trained neural net would take a few milliseconds to generate the flow field.



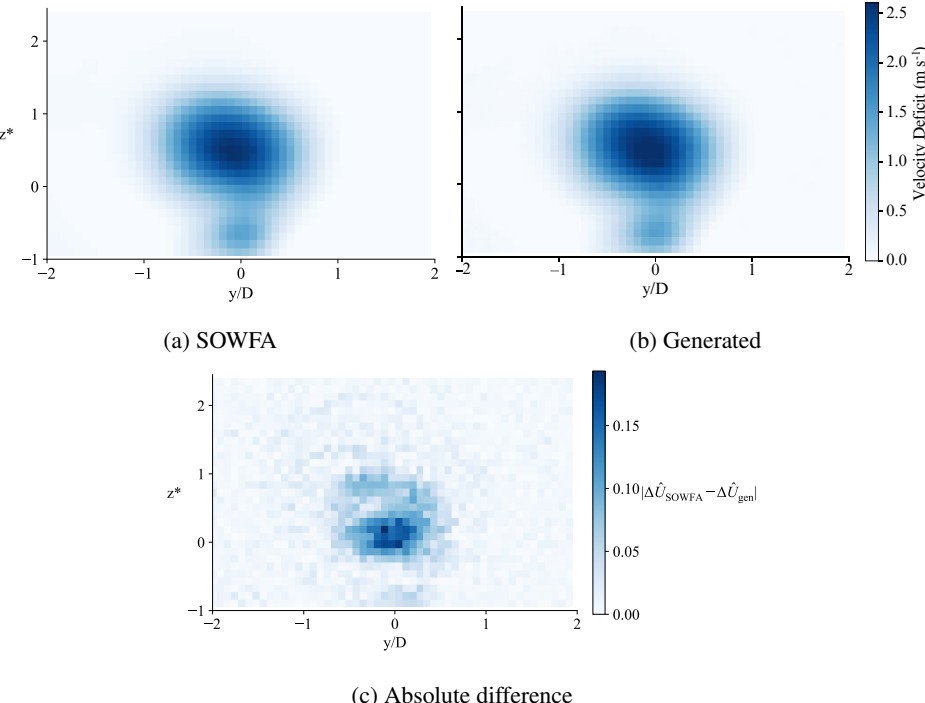

(a) SOWFA

(b) Generated

(c) Absolute difference

**Figure 16.** SOWFA data image (a) compared to the neural network's generated image (b) at 7.5 x/D of a turbine tilted -20°. The absolute difference between these images is displayed in (c).

## 4  Comparison of results

In this section, the results of each wake modeling approach are compared and contrasted in order to understand the benefits and constraints of each approach. Figure 17 visually compares the ability of the deep learning approach (see Fig. 17b) and the optimized Bastankhah wake model (see Fig. 17c) to accurately predict the cross-stream velocity profile at 12.0 rotor diameters downstream when the upstream turbine is tilted 7.5°. According to Fig. 11, the optimized wake model approach greatly reduced the RMS error at 12.0 rotor diameters downstream with 7.5° of tilt. However, the trained neural net is able to generate essentially identical results to SOWFA.

Although the optimization approach to calibrating the Bastankhah wake model does not perform as well as the deep learning approach, there are benefits to the optimization approach. The optimized Bastankhah wake model can be used in various wind farm optimization tools without significant changes to existing workflows. Thus, this approach is useful in calibrating existing and future additions and modifications to wake models used in wind farm optimization workflows. Figure 18 reveals physically where the optimized coefficients improved accuracy in the velocity field prediction. These cross-sections are the sum of $|\Delta \hat{U}|^2$ for each point in the cross-stream slices over all the tilt angles and downstream distances used in the optimization. $|\Delta \hat{U}|^2 = |\hat{U} - \hat{U}m|^2$, where $\hat{U}$ is the SOWFA stream-wise velocity deficit and $\hat{U}_m$ is the velocity deficit predicted by the





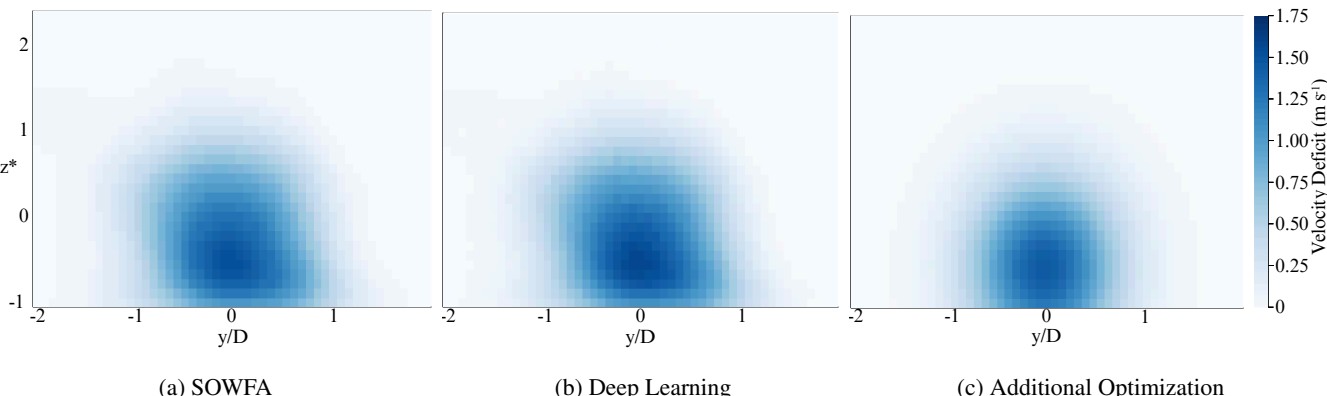

**Figure 17.** Comparison of the deep learning and additional optimization approach to predicting the velocity deficit of the wake of a $7.5°$ tilted turbine at 12.0 rotor diameters downstream.

modified Bastankhah wake model. The most reduction in error occurred in the area around $-0.5$ y/D and $\pm0.5$ z*. There is a
slight increase in error in part of the wake using the coefficients from the additional optimization step, as can be seen in the area around $0.5$ y/D and $-0.6$ z*. Overall, the additional optimization step improved the model's ability to accurately represent the tilted wake.

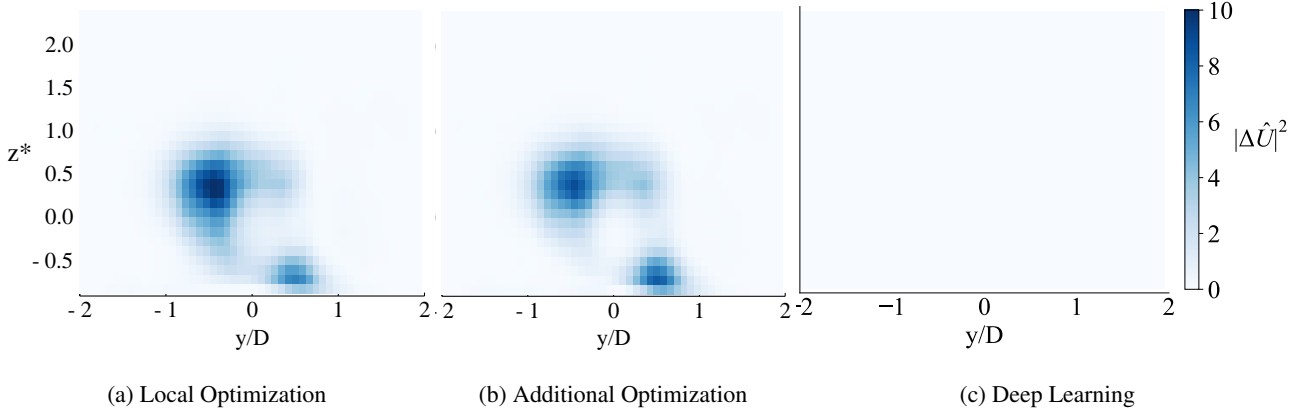

**Figure 18.** Sum of $|\Delta\hat{U}|^2$ for each point in the cross-stream slices over all the tilt angles and downstream distances used in the optimization. $|\Delta\hat{U}|^2 = |\hat{U} - \hat{U}m|^2$, where $\hat{U}$ is the SOWFA stream-wise velocity deficit and $\hat{U}_m$ is the velocity deficit predicted by the modified Bastankhah wake model (a and b) and the deep learning model (c). (b) uses the optimized coefficients to predict $\hat{U}_m$, and (a) uses the locally optimized coefficients.

Figure 16c shows that the main inaccuracy in the neural net predictions is near the center of the wake; however, this error is insignificant when compared to the $|\Delta\hat{U}|^2$ of the optimized Bastankhah wake model (see Fig. 18). Compared to the perfor-





mance of the local optimization and additional optimization approaches, the deep learning approach has no perceivable error (see Fig. 18c).

## 5    Summary and future work

This comparative analysis of wake modeling approaches for tilted wind turbines underscores the advancements and effectiveness of both traditional optimization methods and deep learning techniques. The process of optimizing the modified Bastankhah
wake model, which involved refining its coefficients to better align with empirical relationships for wake behavior, significantly improved the model's accuracy. This optimization process focused on calibrating the model to account for the tilt angle, deflection, and wake growth, effectively reducing the RMS error by about 15% within a defined range of tilt angles.

The optimization approach entailed an extensive analysis of the relationships between tilt angle, wake deflection, and wake growth. By calibrating the coefficients of the modified Bastankhah model, the approach addressed some of the inaccuracies
associated with the original model's assumptions about wake shape. This method yielded a notable reduction in RMS error and demonstrated an improved representation of wake dynamics. However, this approach remained limited by the constraints of a Gaussian wake description and the need for significant computational resources and time for analysis.

In contrast, the introduction of deep learning techniques marks a significant advancement in wake modeling. A neural network was designed with four layers, featuring batch normalization and ReLU activation functions, and was expanded from
an initial input layer to a final output of 31x51 pixels. This architecture enabled the network to capture complex wake structures with high precision. The training involved 2,500 epochs with the SSIM objective followed by 2,000 epochs with the RMS error objective, resulting in a 95% reduction in RMS error compared to the optimized Bastankhah model. The SSIM objective facilitated initial training by reducing noise and capturing the general wake shape, while subsequent RMS error training refined the model to generate more accurate wake structures.

The deep learning model's performance, which produced wake predictions in milliseconds as opposed to the 10 hours required for SOWFA simulations, highlights its efficiency and practical applicability. The model's ability to closely match SOWFA data, with only minor inaccuracies near the center of the wake, demonstrates its robustness in handling complex wake phenomena.

While the optimized Bastankhah model continues to offer value, particularly in integrating with existing wind farm opti-
mization tools, the deep learning approach provides a more precise prediction. This advancement offers significant potential for enhancing wake prediction accuracy and efficiency for floating offshore wind farms, where the movement and size of the turbines develop unsteady wakes that cannot be accurately described by traditional wake models. Future research should aim to further refine neural network performance and explore its broader application across diverse wind farm scenarios.

*Code and data availability.*   https://github.com/byuflowlab/cutler-2024-tilted-wake.git



*Author contributions.* JC led this research, including developing and comparing the wake models, running the deep learning and optimizations, and writing the paper. CB set up and ran SOWFA to obtain the high-fidelity flow field data and helped develop ideas. AN helped develop ideas and methodology, provided feedback throughout the entire process, and provided editing for the paper.

*Competing interests.* The contact author has declared that neither they nor their co-authors have any competing interests.





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
