# Peer review of "Introduction and comparison of deep learning and optimization approaches to analytical wake modeling of a tilted wind turbine"

_Wind Energy Science, 2024_

## Author Comment (AC1)

I.  RESPONSES TO RC1

   A.  The paper presents an analysis and different methods to predict the wake behavior of tilted wind turbines. The model proposed by Bastankhah and Porte-Agel for wind turbines in yaw is extended to tilted wind turbines. A deep learning approach is proposed as an alternative to solving the flow equations to calculate the detailed wake structure. The paper contains relevant information and indicates interesting approaches to study the wakes of tilted wind turbines. Nevertheless, there are significant aspects of the manuscript that require clarification.

   B.  As stated in line 68, the methodology proposed has been obtained for some specific conditions and is not generalizable to other ones. Nevertheless, a justification of why these conditions have been chosen is of interest, do they correspond to the more usual or representative working conditions?

      1.  Yes, these conditions are fairly representative of normal working conditions (wind speed of 8.0 m/s and turbulence intensity of 0.08). I have now included the following sentence:

         "The additions and adjustments were calibrated on data representative of normal working conditions with a wind speed of 8.0 m/s and a turbulence intensity of 0.08"

   C.  Besides, these working conditions should be more clearly specified; some information is given in line 100, but I miss other relevant parameters dealing with wakes, like the thrust coefficient, ambient turbulence and other inflow conditions. Also, the main machine characteristics and dimensions should be also included without needing to consult bibliography.

      1.  Good catch - this needs to be clarified for reproducibility. I have now included the relevant parameters used. However, the physical dimensions of the NREL 5MW can be lengthy to include. I have included the hub-height and the rotor-diameter, however, for more detailed dimensions the reader can see the citation. Here is the portion of the paper that I have edited to include these details:

         "A 5-MW NREL reference turbine was simulated in SOWFA over varying degrees of tilt at a wind speed of 8 m/s, low turbulence intensity of 0.08, coefficient of thrust (CT ) of 0.8, shear of 0.15, and a neutral atmospheric boundary layer (Churchfield et al.105 (2012)). The flow field results were averaged over the run time of 2,500 seconds where the flow converged. The turbine hub-height was set to 90.0 meters with a rotor diameter of 126.0 meters."

D. It is not clear what data are you using to train the additional optimization and deep learning approaches, and what data are you using for validation and checking the results. A similar comment can be made about the surrogate model of vertical deflection. A brief comment is made in line 285, but it is not clearly justified if the training and validation data sets, both belonging to the same working conditions, are really independent.

    1. I have included more details of the data used to train the additional optimization and deep learning approaches. The additional optimization uses the same training data as the local optimization approach. The local optimization approach helps define the required empirical additions and adjustments and then the additional optimization uses the same data to further calibrate the parameters of the newly defined empirical additions and adjustments. The deep learning approach uses the same dataset as the local and additional optimization approaches including additional data for additional turbine tilt angles because the deep learning approach isn't limited to a range of tilt angles. Here is the revision in the paper:

    "In order to thoroughly train our neural net, we used 1,850 cross-stream slices from the SOWFA data velocity field over varying tilt angles ranging from -35∘ to 25∘. The SOWFA data used holds the same turbine characteristics and flow field conditions as the data used to define the empirical relationships and implement the additional optimization step for the modified Bastankhah wake model. The 1,850 images were then randomized into separate training and validation datasets using PyTorch's randperm function, which implements the Mersenne Twister pseudorandom number generator (Imambi et al. (2021)). Although this data set is split into training and validation datasets it does not mean this model is generalizable. In future work, a more expansive training and validation data set that spans varying turbine types and flow field conditions would enable the model to be generalizable. However, for the purpose of comparing different approaches of analytical wake modeling we have limited the training and validation dataset to the working conditions detailed in section 2.1."

E. Besides, the usefulness of the proposed models is not clear, as you have to solve SOWFA first to get the training data for this particular situation. It may be that if in future work you are contemplating several different working conditions, the utility of the method would be more patent.

    1. It is true that the immediate usefulness of these models is limited due to the models being calibrated and trained on this particular situation. The

main purpose of comparing these models is to identify limitations and benefits of each approach. Future work can certainly include developing these models for a broader range of working conditions. However, we feel it is important to first understand which modeling approaches are more promising for developing complex wake modeling capabilities especially for floating offshore wind farms that deal with complex wakes due to the movement of the floating platform.

F. I think that the Bastankhah and Porte-Agel model was originally proposed for yawed wind turbines and its application to tilted wind turbines is not straightforward, and requires more than an improvement or modification, as seems to be suggested in the abstract and other parts of the paper.

1. Good point - I have now included additional citations that address examples of previous additions and modifications to the Bastankhah and Porte-Agel model to justify our approach in the following portion of text:

   "The Bastankhah wake model has undergone several additions and modifications to account for varying yaw angles and turbulence intensities (Niayifar and Porté-Agel (2016); Bastankhah and Porté-Agel (2016)). This study introduces an improvement to the current approach of building the capabilities of the Bastankhah wake model (2016) as well as a novel deep learning approach to modeling complex wake dynamics."

2. When the Bastankhah wake model was modified and adjusted to include yaw it also acknowledged limitations due to a kidney bean wake shape for large yaw angles. Our approach follows their derivation for wake deflection due to turbine rotor alignment but applies it in the vertical direction. Without the ground, the vertical deflection would follow trajectories similar to yaw deflection; however the presence of ground highly influences the wake recovery, wake deflection, and wake growth. Our local optimization approach was meant to demonstrate an approach used in the past to similarly add to and modify the Bastankhah wake model in order to expand its wake modeling capabilities.

G. In line 103, "SOWFA simulations confirm similar trends to previous studies of tilted turbines..." give references of these previous studies.

1. I have now provided these references:

   "Overall, the results of the SOWFA simulations confirm similar trends to previous studies of tilted turbines (Annoni et al. (2017); Johlas et al. (2022); Bay et al. (2019))."

H.  Figures 1a and 1b opposite of indicated in text.
    1.  Thank you for catching that. I have not corrected it.
I.  In figure 1 and following ones, it is difficult to see the contrast.
    1.  I selected blue shades to ensure that the key aspects of the figures remain clear, even when printed in black and white. The focus is on observing the overall shape rather than analyzing specific velocity deficit values.
J.  It is not clear how figure 2a is obtained. Bastankhah and Porte-Agel 2016 is for yawed wind turbines.
    1.  I have now included in the description of Figure 2 that the stream-wise slice comes from a tilted turbine simulated in SOWFA.

        "Stream-wise slice of the velocity deficit centered at a 2.5∘ tilted turbine (simulated in SOWFA). z* represents the vertical position normalized with the hub height (90 meters)"

K.  Figure 2b is not referred in the text
    1.  Good catch - I have now removed figure 2b, and there are other figures that can be referenced instead of 2b.
L.  Regarding line 114, cross stream slices should be symmetrical, but frequently they are not because of unavoidable errors. How do you deal with this asymmetry?
    1.  This is a major limitation of the local and additional optimization approaches. They both assume the cross stream slice of the wake can be sufficiently estimated with a symmetrical shape. However, the local and additional optimization methods both account for vertical and horizontal deflection. Section 2.1.1 goes into how we account for some of the asymmetry.
M.  In the caption of figure 3, how are the solid lines obtained, also from SOWFA?
    1.  Good catch - the solid lines and points are both from SOWFA but the points assume the center of the wake moves vertically and horizontally and the solid lines assume the wake only deflects vertically. I have now clarified in the figure description that both the solid lines and dotted lines come from SOWFA data:

        "Deflection of tilted turbine wakes as observed from cross-stream slices in SOWFA (marked with points) and the deflection observed from a stream-wise slice in SOWFA (marked with solid lines)."

N.  In lines 141 to 145, text not very clear.
    1.  I have revised lines 141 to 145 to the following:

"In order to focus on accurately estimating the upper portion of the vertical velocity profile the profile is split at the point of max velocity deficit, at the peak of the gaussian shape (see Fig. 4). Then the upper portion of the velocity profile is mirrored across the point. For example, observing Figure 4, this would entail removing the portion of the SOWFA data that is less than a $z*$ value of around 0.81. Then mirroring the remaining SOWFA data across $z* = 0.81$. This forms a normal Gaussian shape where a normal Gaussian fit is used to find $\sigma z$."

---

## Author Comment (AC2)

I. RESPONSES TO RC2

A. At present, only one inflow condition is investigated. It therefore seems to be more of a proof of concept or case study since it is not clear to what degree the results are applicable to different ambient conditions, in particular since you argue in ll. 271 that the Bastankhah model could only be improved if more parameters would be included. If this is the intention, clarify.

1. Yes, this is more of a proof of concept in order to compare and contrast the different wake modeling approaches. I have edited the sections that address the dataset to clarify that these approaches have potential to be generalized over varying conditions, but for this paper we are only examining the wake of one turbine with one inflow condition and varying tilt angles.

"In order to thoroughly train our neural net, we used 1,850 cross-stream slices from the SOWFA data velocity field over varying tilt angles ranging from -35∘ to 25∘. The SOWFA data used holds the same turbine characteristics and flow field conditions as the data used to define the empirical relationships and implement the additional optimization step for the modified Bastankhah wake model. The 1,850 images were then randomized into separate training and validation datasets using PyTorch's randperm function, which implements the Mersenne Twister pseudorandom number generator (Imambi et al. (2021)). Although this data set is split into training and validation datasets it does not mean this model is generalizable. In future work, a more expansive training and validation data set that spans varying turbine types and flow field conditions would enable the model to be generalizable. However, for the purpose of comparing different approaches of analytical wake modeling we have limited the training and validation dataset to the working conditions detailed in section 2.1."

B. More details have to be given on the training data:

Details of the turbine operational state (thrust coefficient)

Details of the inflow: TI, shear, veer; is this an atmospheric boundary layer inflow or a uniform inflow? It the latter: why? If the former: What was the atmospheric stability?

The data seems to be averaged. Is the average converged? Over what time period/how many steps was the average calculated?

1. We used a thrust coefficient of 0.8

2. We used a turbulence intensity of 0.08, shear is set at 0.15, and this is atmospheric boundary layer inflow. With the turbulence intensity at 0.08 and the shear at 0.15 the atmospheric stability was neutral (Ri was approximately 0)

3. The flow field is the averaged flow field. The average was calculated over a run time of 2500.0 seconds and it converged.

4. These are all good suggestions of information missing in the paper about the training data. I have included these details in the final draft in the following paragraph:

   "A 5-MW NREL reference turbine was simulated in SOWFA over varying degrees of tilt at a wind speed of 8 m/s, low turbulence intensity of 0.08, coefficient of thrust (CT ) of 0.8, shear of 0.15, and a neutral atmospheric boundary layer (Churchfield et al.105 (2012)). The flow field results were averaged over the run time of 2,500 seconds where the flow converged. The turbine hub-height was set to 90.0 meters with a rotor diameter of 126.0 meters."

C. It is not clear how you generate the training data and what data you use to compare your results to. Is it the same data for both?

1. Yes, the training data used in the additional optimization model and the deep learning model is the same SOWFA data used to calibrate the locally optimized Bastankhah wake model. When training the deep learning model the data was randomly split into a training and validation dataset. I have now clarified this in the paper when I detail the training data. It is in the same revised section that I mentioned above, but I will copy it here again for simplicity:

   "In order to thoroughly train our neural net, we used 1,850 cross-stream slices from the SOWFA data velocity field over varying tilt angles ranging from -35∘ to 25∘. The SOWFA data used holds the same turbine characteristics and flow field conditions as the data used to define the empirical relationships and implement the additional optimization step for the modified Bastankhah wake model. The 1,850 images were then randomized into separate training and validation datasets using PyTorch's randperm function, which implements the Mersenne Twister pseudorandom number generator (Imambi et al. (2021)). Although this data set is split into training and validation datasets it does not mean this model is generalizable. In future work, a more expansive training and

validation data set that spans varying turbine types and flow field conditions would enable the model to be generalizable. However, for the purpose of comparing different approaches of analytical wake modeling we have limited the training and validation dataset to the working conditions detailed in section 2.1."

D. There have been other attempts at modeling tilt, more context and motivation of why you chose the Bastankhah model should be added in the introduction.
    1. I have included more details of the motivation behind using the Bastankhah wake model in the introduction (see line 30 in the revised draft)

      "The Bastankhah wake model has undergone several additions and modifications to account for varying yaw angles and turbulence intensities (Niayifar and Porté-Agel (2016); Bastankhah and Porté-Agel (2016)). This study introduces an improvement to the current approach of building the Bastankhah wake model's capabilities as well as a novel deep learning approach to modeling complex wake dynamics."

E. 22: there might be more and older works on wake steering.
    1. Good point - I have now included a few more references of older works on wake effects in wind farms.

      "Wind farms lose between 15% and 20% of energy production for a typical wind farm throughout the year because of wake interference between turbines (Barthelmie et al. (2007); Briggs (2013); Barthelmie et al. (2009); Barthelmie and Jensen (2010); Jensen (1983); Voutsinas et al. (1990))."

F. 24 "Bastankhah and Porté-Agel (2016) developed an analytical wake model that is capable of sufficiently modeling the horizontal deflection in the wake" – "sufficiently" is rather unspecific.
    1. I have changed it to "Bastankhah and Porté-Agel (2016) developed an analytical wake model that is capable of accurately modeling the horizontal deflection in the wake"

G. 29 – combinations of tilt and yaw will be even more complex. In general, you need to specify which version of the Bastankhah model you refer to (2014 or 2016) - maybe call them Bastankhah 2014/Bastankhah 2016 wake model
    1. I have now specified we are using the Bastankhah 2016 model.

"This study introduces an improvement to the current approach of building the capabilities of the Bastankhah wake model (2016) as well as a novel deep learning approach to modeling complex wake dynamics. The improvements and limitations of these approaches are demonstrated in the modeling of a tilted wind turbine's wake for various tilt angles."

H. It should be specified in which direction the wake is deflected based on which direction the wind turbine is tilted to (e.g. l. 45, 102)
   1. I have now included the following description in the introduction:

   "Throughout this paper the tilt angles specified can be assumed to represent an initial wake deflection angle of equal magnitude in the opposite direction. For example, 15 degrees of tilt would results in an initial wake deflection angle of -15 degrees"

I. 56 "holds" (not hold, because it refers to range)
   1. Good catch - It has been fixed.

   "Based on the analysis of cross-stream slices, we can define a range of tilt angles that holds the assumption of a Gaussian distribution description as well as any necessary additional empirical relationships for variables dependent on tilt."

J. 72: which tilt angles would be expected for floating offshore wind turbines?
   1. For normal operating conditions it reaches up to 8 degrees, but if it is controlled then it could be around 10-15 degrees. It depends on the floating structure.

K. 81: how is your approach different that you are able taking into account the full wake?
   1. It is important to be able to resolve the cross-stream slices at downstream locations in order to accurately estimate the wake effect on downstream turbines.

L. Figure 1: what causes the span-wise asymmetry of the wake? How was the wake center determined?
   1. The span-wise asymmetry is mainly caused by the proximity of the wake to the ground. The center of the wake was determined by conducting a 2d-interpolation of the span-wise slice and then resolving the data to a higher resolution and finding the point of maximum deficit.

M. 113 (from "Observing…"): Rephrase the sentence, it is difficult to understand what you mean

1. Here is the revision:

   "Observing downstream vertical velocity profiles based on a stream-wise slice of the flow field assumes that there is insignificant horizontal deflection of the wake center. However, for tilted turbines there is both horizontal and vertical deflection which displaces the center of the wake out of the stream-wise plane. Thus, the observed downstream vertical velocity profiles are inaccurate as they are not aligned with the true wake center."

N. The blue shades are sometimes hard to distinguish
   1. I chose blue shades so that, even if it was printed in black and white, the main point of the figures would come through. Since the point of the figures is to observe the general shape rather than examine the velocity deficit values.
O. 121: k is called "wake growth rate" in Bastankhah 2016.
   1. It's been corrected in my paper now.

      "Equations 1 and 2 define $\sigma y$ and $\sigma z$ as having a linear relationship with the downstream distance ( $x-x0d$ ) where the slope ($ky$ and $kz$ ), or commonly referred to as the wake growth rate, is determined by applying a linear fit to $\sigma y$ and $\sigma z$"

P. Section 2.1.1: At present, I do not see a reason for this detailed investigation - what is the aim here? To show that $\sigma$ can be determined up to 12.5 degrees from mirroring the top half of the skewed Gaussian profile?
   1. That is the main purpose of this section. Although it can be lengthy, it was to point out the process for determining the 12.5 degree limit.
Q. Section 2.1.2: this subsection has 4 lines of text and one figure. Together with 2.1.1, this could be summarized to briefly illustrate the change in wake shape when crossing 15 degrees (I assume that this is the information that you want the reader to have?).
   1. That is a good point. I have now combined the sections to bring more clarity to this portion of the paper:

**2.1.1 Model limitations**

There are limitations on the abilities of the Bastankhah wake model to capture complex wake shapes. The main foundational
assumption in the Bastankhah wake model assumes a normal Gaussian shape in the vertical and horizontal velocity profiles.
However, when the wake compresses vertically it forms a skewed Gaussian shape (see Fig. 4). Skewed Gaussian shapes have
also been observed in analyses of yawed turbine wakes; however, the skew was insignificant enough to maintain the assumption
of a normal Gaussian shape (Bastankhah and Porté-Agel (2016)). Similarly, for small positive tilt angles, the skew is negligible
and a normal Gaussian fit sufficiently defines the wake shape and wake growth. However, with the presence of the ground,
the skew can not be ignored for large tilt angles. A skewed Gaussian fit would be better suited to approximate the vertical
velocity profile; however, this would conflict with assumptions used to derive the Bastankhah wake model (Bastankhah and
Porté-Agel (2016)). Although the skew becomes more prominent for large tilt angles, the deflection of the center of the wake
places the bottom portion of the wake away from the rotor swept areas of downstream turbines (assuming the same hub height).
Therefore, a normal Gaussian fit can still be used as long as it accurately approximates the upper portion of the vertical velocity
profile for large tilt angles (see Fig. 4).

[Figure]

**Figure 4.** Vertical velocity profile at 14 x/D downstream of a turbine with 12.5° of tilt.

In order to focus on accurately estimating the upper portion of the vertical velocity profile the profile is split at the point of
max velocity deficit, at the peak of the gaussian shape (see Fig. 4). Then the upper portion of the velocity profile is mirrored
across the point. For example, observing Figure 4, this would entail removing the portion of the SOWFA data that is less than
a $z^*$ value of around 0.81. Then mirroring the remaining SOWFA data across $z^* = 0.81$. This forms a normal Gaussian shape
where a normal Gaussian fit is used to find $\sigma_z$. In order to accurately define the relationship between $\sigma_z$ and tilt, SOWFA
simulations were run for a single turbine at tilt angles of 2.5°, 5°, 7.5°, 10°, and 12.5° (see Fig. 5a). A normal Gaussian fit
without any required mirroring of the data was used to measure $\sigma_y$ (see Fig. 5b). Similar to what has been observed with
turbine yaw, $\sigma_z$ and $\sigma_y$ can be observed to increase linearly with respect to the downstream distance even for larger angles
of tilt. Bastankhah and Porté-Agel (2016) observed that the rate at which $\sigma_y$ increased was constant over varying yaw angle.
However, the rate at which $\sigma_z$ and $\sigma_y$ increase is variable over varying tilt angles. Thus, empirical relationships are necessary
to define the change in slope over varying tilt angles for both $\sigma_z$ and $\sigma_y$.

In addition to the challenges of modeling the skewed wake shape there are limitations on modeling large deformations in
the wake due to large tilt angles. When the turbine is tilted beyond 15°, a kidney bean shape begins to form (see Fig. 6). The
kidney bean shape would require a double Gaussian shape to approximate, whereas the Bastankhah wake model derivation
relies on a single Gaussian shape description (Johlas et al. (2022)). Therefore, the calibrated analytical wake model described
in this paper is bounded to tilt angles less than 15°. This limit is reasonable as it is in line with limits set for fixed offshore
platform tilt (Ramachandran et al. (2017).

R. **Section 3.3: With the current progress in AI and computational power, in your opinion, how long will wake models be necessary before deep learning approaches take over? You clearly show that if you have training data, it is more powerful to use deep learning for generating the wake field than for tuning parameters of a model.**

   1. I think there is plenty of work left to do in implementing deep learning wake models. But with enough training data I feel that sometime within

4-10 years the deep learning wake models will be much more accurate and available.

S. 313 "The optimized Bastankhah wake model can be used in various wind farm optimization tools without significant changes to existing workflows." since the optimization depends on

turbine type
inflow speed / turbine operation
inflow turbulence/atmospheric stability
wind shear and veer this does not seem to be a trivial modification. Please comment.

1. That's a valid point. To clarify I have rephrased this to:

"The optimized Bastankhah wake model seamlessly integrates into various wind farm optimization tools without requiring significant workflow modifications and, when provided with sufficient training data, incorporates more effectively into existing workflows compared to the deep learning wake model under identical training conditions."